# Perceptions of 3R implementation in European animal research: A systematic review, meta-analysis, and meta-synthesis of barriers and facilitators

Edwin Louis-Maerten[ID][1]*, Aoife Milford[1], David M. Shaw[1,2], Lester D. Geneviève[1], Bernice S. Elger[1,3]

**1** Institute for Biomedical Ethics, University of Basel, Basel, Switzerland, **2** Care and Public Health Research Institute, Maastricht University, Maastricht, The Netherlands, **3** Faculty of Medicine, Center of Legal Medicine, University of Geneva, Geneva, Switzerland

* edwin.louis@unibas.ch

## Abstract

### Objectives

The purpose of this systematic review was to examine how the scientific community in Europe that is involved with research with animals perceives and experiences the implementation of 3R (Replace, Reduce, Refine).

### Methods

A systematic search of the literature published in the past ten years was performed in PubMed, Web of Science and Scopus. Publications were screened for eligibility using *a priori* inclusion criteria, and only empirical evidence (quantitative, qualitative, or mixed methodologies) was retained. Quantitative survey items were investigated by conducting a meta-analysis, and the qualitative data was summarized using an inductive meta-synthetic approach. Included publications were assessed using the Quality Assessment for Diverse Studies tool.

### Results

17 publications were included (eight quantitative, seven qualitative, two mixed-methods). The meta-analysis revealed that scientists are skeptical about achieving replacement, even if they believe that 3R improve the quality of experimental results. They are optimistic concerning the impact of 3R on research costs and innovation, and see education as highly valuable for the implementation of 3R. The meta-synthesis revealed four barriers (systemic dynamics, reification process, practical issues, insufficient knowledge) and four facilitators (efficient use of animals, caring for animals, regulatory uptake, supportive workplace environment).

### Conclusion

These findings show actionable levers at the local and systemic levels, and may inform regulators and institutions in their 3R policies.

**Data Availability Statement:** All relevant data are within the manuscript and its Supporting information files.

**Funding:** The authors acknowledge the support of the Swiss National Science Foundation (SNSF National Research Program (NRP) - 79 Advancing 3R – Animals, Research and Society, grant number 206432) in funding this research.

**Competing interests:** The authors have declared that no competing interests exist.

## Trial registration

The protocol was registered into the PROSPERO database under the number CRD42023395769.

## 1. Introduction

In their *Principles of Humane Experimental Technique*, W. M. S. Russell and R. L. Burch introduced the Three R principles (replacement, reduction and refinement; hereafter 3R) in animal experimentation [1], which aimed to render experimental procedures on nonhuman animals more morally acceptable. They defined *replacement* as the use of "any scientific method employing non-sentient material", *reduction* as minimizing, other than by replacement, "the number of animals used to obtain information of a given amount and precision", and *refinement* as a "decrease in the incidence or severity of inhuman procedures applied to those animals which have to be used" [1, p.64]. Although these definitions have been criticized for perceived vagueness and incompleteness [2–8], 3R has gained popularity during the past decades and it has become a cornerstone in debates on animal research. In addition, their implementation in local, national and international regulations can be seen as a significant milestone regarding the ethical use of nonhuman animals in research and education. For instance, the European Directive 2010/63/EU explicitly states that:

> "The care and use of live animals for scientific purposes is governed by internationally established principles of replacement, reduction and refinement. [...] [T]he principles of replacement, reduction and refinement should be considered systematically when implementing this Directive. [...] [T]his Directive represents an important step towards achieving the final goal of full replacement of procedures on live animals for scientific and educational purposes as soon as it is scientifically possible to do so."

[9, p.34].

However, some reports suggest that the worldwide implementation of the 3Rs is not as efficient as it should be, in particular when it comes to promote replacement, and more specifically a goal of full replacement [4,10–12]. Many reasons may be invoked such as inoperative policies, inadequate funding, or some forms of inertia in changing research procedures [13]. As of yet, there is no appraisal of the existing literature to assess the actual impact of each of these reasons on the implementation of the 3Rs from the perspective of the animal research community. A preliminary search of two online databases (PubMed and Web of Science), which was undertaken by the authors of the present review, showed the existence of empirical publications exploring the perspectives of various stakeholders from the field of animal experimentation on the implementation of the 3Rs. The present systematic review is aimed at identifying barriers and facilitators experienced by scientists (in a broad sense, see next section for details) in their implementation of 3R practices. Therefore, conducting a review of these empirical publications may shed light on factors influencing the implementation of 3R, including existing best practices and possible supportive 3R measures that can readily be applied. It would also provide a more transparent understanding of the actual implementation of the 3Rs to policymakers, animal welfare bodies, governmental and non-governmental organizations, as well as the animal research community itself.

The objective of this review is to examine the impact of 3R implementation on the experiences and perceptions of European scientists conducting research with animals. Many attitudes have been assessed by previous studies either by using a quantitative approach (*e.g.* surveys with closed-ended questions) or with a qualitative approach (*e.g.* interviews with open-ended questions). Thus, in order to address the review objective, quantitative, qualitative, and mixed-methods publications were considered. A quantitative and qualitative synthesis of all data gathered was also conducted in order to provide reliable conclusions with this review.

## 2. Results

### 2.1. Search results

In total, 2,328 publications were identified through the search in online databases. 1,345 publications were screened for relevance at the title and abstract level after the removal of duplicates. Most publications were irrelevant for this review, making 65 publications suitable for full-text screening. From these, 49 publications were excluded either because their design was not empirical (n = 22), they did not address the experiences or perceptions towards the implementation of 3R (n = 21), they did not involve scientists (n = 1), they did not address the context of research on animals (n = 1), or because we were unable to retrieve the full text (n = 4). After screening the references of all included publications, we identified 1 additional publication meeting the inclusion criteria. In the end, 17 publications were included for review (Fig 1): eight of them had only quantitative data, seven had only qualitative data, and two had

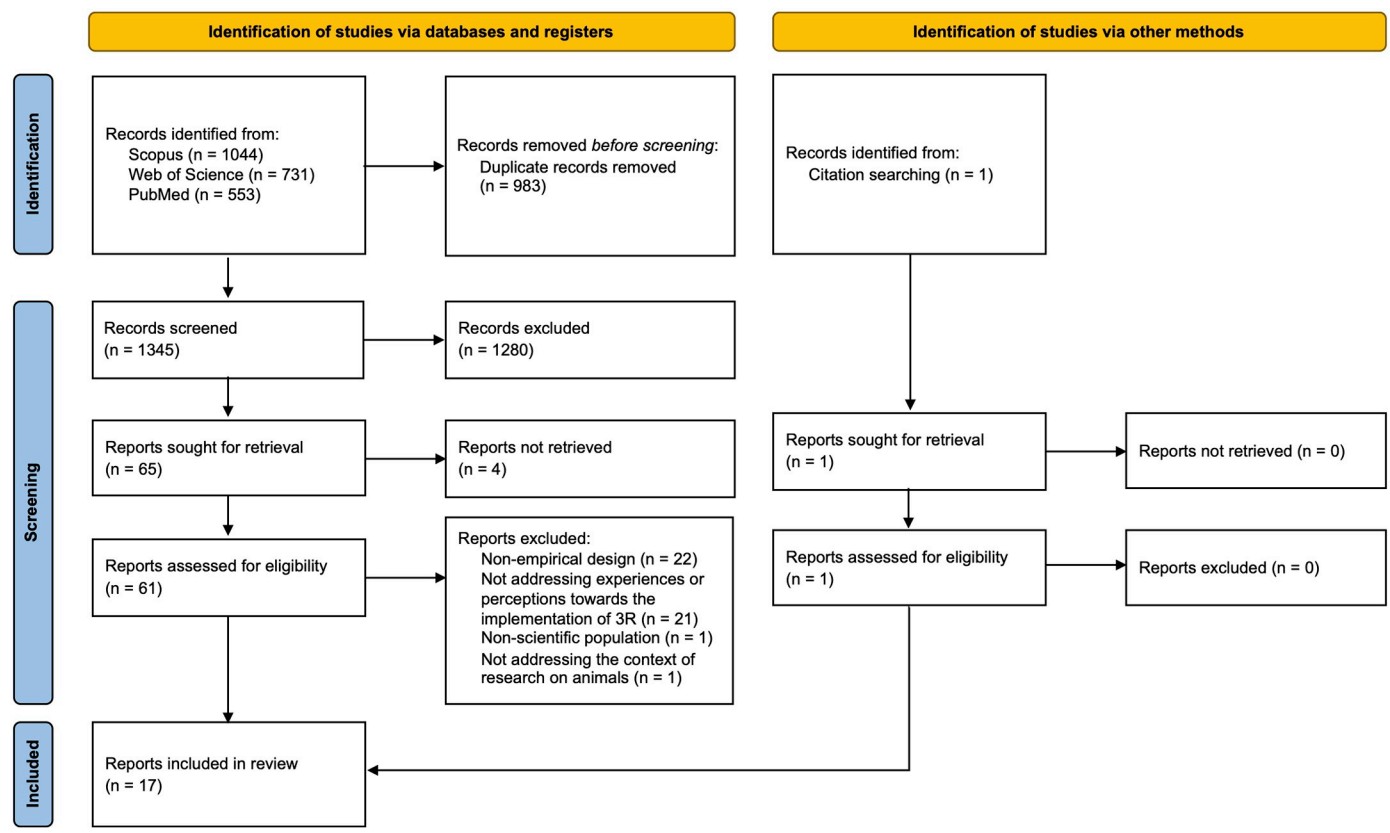

**Fig 1. PRISMA flow diagram for systematic search and selection protocol (adapted from [14]).**

mixed methodologies. The publications represented different populations of scientists. All publications with quantitative data employed a survey design, whereas publications that were purely qualitative had more various designs. Table 1 provides a description of the included publications.

**Table 1. Description of included publications.**

| Publication | Methodology (design) | Countries included | Population | Number of participants |
|---|---|---|---|---|
| Baldelli 2019 [15] | Quantitative (survey) | Italy | Students | 733 |
| Bressers 2019 [16] | Quantitative (survey) | The Netherlands | Researchers | 367 |
| Edwards 2015 [17] | Quantitative (survey) | United Kingdom | Researchers | 59 |
| Franco 2014 [18] | Quantitative (survey) | Portugal | Researchers | 206 |
| Franco 2018 [12] | Quantitative (survey) | Portugal Germany Switzerland Denmark | Researchers | 310 |
| Lindsjö 2021 [19] | Quantitative (survey) | Sweden | Animal Welfare Officers Veterinarians Researchers Animal technologists Ethologists | 44 |
| Nøhr 2016 [20] | Quantitative (survey) | Denmark | Researchers | 234 |
| Varoni 2023 [21] | Quantitative (survey) | Italy | Students | 501 |
| Crettaz von Roten 2018 [22] | Mixed-methods (survey) | Switzerland | Animal technologists Students Researchers | 510 |
| Van Luijk 2013 [23] | Mixed-methods (survey) | The Netherlands | Animal Welfare Officers | 15 |
| Brønstad 2019 [24] | Qualitative (focus groups) | The Netherlands Denmark Sweden Norway | Researchers Veterinarians Animal technologists Bioethical assistant | 21 |
| Cabaret 2022 [25] | Qualitative (interviews) | France | Researchers Animal technologists Animal caretakers Abattoir worker | 12 |
| Greenhough 2018 [26] | Qualitative (ethnographic study) | United Kingdom | Animal technologists Animal facility directors Members of the Institute of Animal Technology and Animals in Science Education Trust Members of NGOs | 27 |
| Mazhary 2019 [27] | Qualitative (interviews) | United Kingdom | Animal facility directors Animal technologists Animal welfare officers Veterinarians Researchers Animal Welfare and Ethical Review Body chairs | 37 |
| Message 2019 [28] | Qualitative (interviews) | United Kingdom | Researchers, Animal technologists Animal facility directors Veterinarians Representatives of animal welfare charities Regulators | 27 |
| Veening-Griffioen 2020 [29] | Qualitative (content analysis) | The Netherlands | Full-text animal research project applications | 110 |
| Williams 2021 [30] | Qualitative (focus groups) | United Kingdom | Animal technologists Researchers | 22 |

## 2.2. Quality assessment

Quality appraisal scores were ranging from 14/39 to 34/39, and the mean and median scores from the two review authors were 24/39 and 25/39 respectively. Therefore, the overall quality of the included publications was considered as satisfactory for analysis.

## 2.3. Meta-analysis

Table 2 provides the list of the 25 quantitative survey items assessed with their respective publications and proportions. Out of these, six survey items were only investigated in one publication and were therefore not considered for meta-analysis (Q1, Q5, Q9, Q10, Q11, Q24). S4 Appendix includes the meta-analysis and associated forest plots of all 19 remaining survey items. A general finding concerning the analyses was their surprisingly low to moderate heterogeneity with $I^2$ statistics values ranging from 0% to 49% (median = 46%; mean = 41%). Five of them (Q6, Q7, Q21, Q23 and Q25) showed presence of an outlying study (*i.e.* a study falling outside the 95% CI and thus likely to skew the results) and were thus reanalyzed without it (the sign "*" will be added hereafter for clarity). The (re)analyzed survey items could be divided into three groups:

1. survey items with no noticeable pooled effect (Q2, Q4, Q12, Q15, Q17, Q20, Q22).

2. survey items that demonstrate a pronounced pooled effect, but with inconclusive prediction intervals (Q3, Q13, Q14, Q16, Q19, Q25*).

3. survey items that demonstrate a pronounced pooled effect with interpretable prediction intervals (Q6*, Q7*, Q8, Q18, Q21*, Q23*).

Concerning group (2), the absence of a conclusive prediction interval can mainly be explained by the small number of publications included (two publications in Q3, Q13, Q14, Q16 and three publications in Q19). Due to space constraints, only survey items from groups (2) and (3) are reported, and the forest plots are presented in S4 Appendix.

The proportion of agreement to some perceptions about the implementation of the 3Rs (Q3, Q6*, Q7*) and four possible facilitators (Q13, Q16, Q18, Q19) tended to have a pooled effect ranging from fairly high for Q19 (67.1%, 95% CI 59–75%) to almost a consensus for Q18 (90.9%, 95% CI 85–95%). Contrariwise, the proportion of agreement to one perception about the implementation of the 3Rs (Q8), one possible facilitator (Q14) and three possible barriers (Q21*, Q23*, Q25*) tended to have a pooled effect ranging from very low for Q23* (6.5%, 95% CI 4–10%) to intermediate for Q8 (20.7%, 95% CI 16–27%).

## 2.4. Meta-synthesis of qualitative data

**2.4.1. General findings.**   From the nine included publications, 36 qualitative codes were generated, each representing a specific perception or attitude towards the implementation of 3R. Among these, 20 were categorized as "barriers" and 16 as "facilitators". The codes were assembled into eight overarching themes (four barriers, four facilitators) and their findings were synthesized. Table 3 provides a description of the final codes and themes.

**2.4.2. Barrier 1: Systemic dynamics within animal experimentation.**   Several publications shed light on power imbalances as a systemic dynamic inherent in animal research settings that shapes the decision-making process as well as ethical considerations. Two kinds of power imbalances are most notably reported: seniority and occupation. Indeed, junior-level workers (animal technologists or researchers) often expressed a sense of powerlessness, being hesitant to voice their concerns for fear of repercussions or crossing boundaries [26,30].

**Table 2. Overview of quantitative survey items.**

| Survey item | Publications included | Original survey item (%agreement; %neutral; %disagreement) |
|---|---|---|
| *3R perceptions* | | |
| Q1: 3R implementation is easy | Van Luijk 2013 [23] | "3R implementation is easy" (00; 47; 53) |
| Q2: Existing 3R possibilities are currently optimally applied | Bressers 2019 [16] | "Lack of awareness is a roadblock for implementation and development of alternatives" (25; 06; 69) |
| | Franco 2018 [12] | "My research protocols have sufficient consideration for the 3Rs" (70; 28; 02) |
| | Van Luijk 2013 [23] | "Existing 3R possibilities are optimally applied" (27; 40; 33) |
| Q3: 3R implementation is important for animal welfare | Van Luijk 2013 [23] | "3R implementation is important for animal welfare" (100; 00; 00) |
| | Varoni 2023 [21] | "How important do you consider refinement for animal welfare" (92; 07; 01) |
| Q4: Refinement is the most important R | Franco 2018 [12] | "Full 'Refinement' of animal experiments is more urgently needed than their Replacement" (62; 27; 11) |
| | Lindsjö 2021 [19] | "Refinement is the most pertinent 3R aspect to consider in research" (36; 00; 64) |
| Q5: Refinement is the most achievable R | Franco 2018 [12] | "Full 'Refinement' of animal experiments is a more readily achievable goal than Full 'Replacement'" (80; 16; 04) |
| Q6: 3R implementation leads to better quality of experimental results | Franco 2018 [12] | "'Refinement' measures are a prerequisite for the quality of animal research" (89; 09; 02) |
| | Lindsjö 2021 [19] | "Implementation of the 3R at your university results in improved research quality" (41; 42; 17) |
| | Nøhr 2016 [20] | "Implementing the 3Rs will be detrimental to the quality of my results" (76; 15; 09) |
| | Van Luijk 2013 [23] | "Better animal welfare leads to better experimental results" (93; 07; 00) |
| | Varoni 2023 [21] | "How important do you consider refinement for experimental results" (89; 09; 02) |
| Q7: Complete replacement will not be achieved in the foreseeable future | Bressers 2019 [16] | "Implementation of animal-free innovation by 2025 is not achievable in my field of research" (71; 15; 14) |
| | Franco 2014 [18] | "How would you predict the relevance of animal experiments in the next 50 years" (99; 00; 01) |
| | Franco 2018 [12] | "Full Replacement of animal experimentation can be achieved in the foreseeable future" (67; 24; 09) |
| | Lindsjö 2021 [19] | "Complete replacement of the use of animals in research and testing will never be achieved" (75; 18; 07) |
| | Nøhr 2016 [20] | "Complete replacement of the use of animals in research and testing will never be achieved" (79; 15; 06) |
| | Varoni 2023 [21] | "Do you think the animal model can be replaced with alternative in vitro and in silico methods" (68; 05; 27) |
| Q8: Replacement is possible for my field of study | Crettaz von Roten 2018 [22] | "A course without live animals would have still enabled me to perform my experiments afterwards to the same level of quality" (25; 00; 75) |
| | Franco 2014 [18] | "How would you classify the present role of animal experiments in your field of research" (12; 00; 88) |
| | Franco 2018 [12] | "How would you classify the relevance of animal experimentation in your own scientific work" (24; 13; 63) |
| | Nøhr 2016 [20] | "What would allow you to continue to achieve your research objectives without using animals" (18; 00; 82) |
| | Varoni 2023 [21] | "Do you think it is important for your profession to practice on animals" (25; 03; 72) |
| Q9: Finding information on 3R is simple | Van Luijk 2013 [23] | "Finding information on 3R methods is simple" (00; 33; 67) |
| Q10: There is enough 3R information available | Van Luijk 2013 [23] | "I am satisfied with the availability of 3R information" (20; 40; 40) |
| *3R facilitators* | | |
| Q11: 3R policy at the university is beneficial for the 3R implementation | Lindsjö 2021 [19] | "The university should implement a 3R policy at the university" (59; 00; 41) |

*(Continued)*

**Table 2.** (Continued)

| Survey item | Publications included | Original survey item (%agreement; %neutral; %disagreement) |
|---|---|---|
| Q12: A dedicated budget is beneficial for 3R implementation | Bressers 2019 [16] | "Research funding is a roadblock for implementation and development of alternatives" (67; 11; 22) |
| | Edwards 2015 [17] | "Increased specific funding for human tissue research programmes would enable researchers to use more human tissue" (56; 00; 44) |
| | Lindsjö 2021 [19] | "The university should implement a budget for 3R implementation" (68; 00; 32) |
| | Nøhr 2016 [20] | "Greater availability of funding for 3Rs research would allow scientists to use fewer animals" (33; 00; 67) |
| Q13: High standards from management/hierarchy is beneficial for 3R implementation | Lindsjö 2021 [19] | "The management should expect the staff to have high standards with respect to animal welfare" (72; 00; 28) |
| | Van Luijk 2013 [23] | "Support at the level of a research department would be a primary focus for gaining a more optimal use of existing 3R knowledge" (67; 00; 33) |
| Q14: Improved regulation is beneficial for 3R implementation | Lindsjö 2021 [19] | "Legislative or regulatory changes would benefit the 3R principles at the university" (20; 00; 80) |
| | Nøhr 2016 [20] | "Changes to legislation would allow scientists to use fewer animals" (12; 00; 88) |
| Q15: Improved work from Ethics Committees is beneficial for 3R implementation | Franco 2014 [18] | "Decisions taken by ethics committees should be binding" (58; 00; 42) |
| | Franco 2018 [12] | "The most effective step to reduce the number of animals used would be to apply more rigorous criteria regarding which projects merit approval" (49; 33; 18) |
| Q16: Dedicated animal specialists are beneficial to 3R implementation | Franco 2014 [18] | "I would like to have support from a laboratory animal scientist or a veterinarian at my institution" (70; 00; 30) |
| | Varoni 2023 [21] | "Do you think that [the presence of specialized figures] can help researchers to carry out good experimentation" (78; 20; 02) |
| Q17: A dedicated literature search service is beneficial for 3R implementation | Nøhr 2016 [20] | "A system for conducting literature searches for replacements would allow me to continue to achieve my research objectives without using animals" (04; 00; 96) |
| | Van Luijk 2013 [23] | "A 3R literature search service for specific research would be a primary focus for gaining a more optimal use of existing 3R knowledge" (60; 00; 40) |
| Q18: Education is beneficial for 3R implementation | Baldelli 2019 [15] | "Do you think that an Animal Bioethics module should be included in your degree course" (93; 01; 06) |
| | Crettaz von Roten 2018 [22] | "A mandatory course on laboratory animal science is beneficial" (94; 03; 03) |
| | Franco 2014 [18] | "Does a laboratory animal science course have any influence on the integration of the 3Rs into your own experiments" (84; 00; 16) |
| | Lindsjö 2021 [19] | "3R awareness, education and training are key factors for successful 3R development at the university" (80; 00; 20) |
| | Varoni 2023 [21] | "How important do you think a course on the use of animal in research is to understand how conducting a high-quality research" (95; 03; 02) |
| Q19: Knowledge exchange is beneficial for 3R implementation | Lindsjö 2021 [19] | "Increased sharing of data or collaboration between research groups help reduce the number of animals used at the university" (63; 00; 37) |
| | Nøhr 2016 [20] | "Increased sharing of data or collaboration between research groups would allow scientists to use fewer animals" (71; 00; 29) |
| | Van Luijk 2013 [23] | "Well-facilitated 3R knowledge exchange between individuals within and between organisations would be a primary focus for gaining a more optimal use of existing 3R knowledge" (53; 00; 47) |
| | | *3R barriers* |
| Q20: 3R implementation decreases appreciation by journals | Bressers 2019 [16] | "Publishing in high-impact journals is a roadblock for implementation and development of alternatives" (63; 09; 28) |
| | Edwards 2015 [17] | "A change in journal publishing policies to more readily publish human tissue-based research would enable researchers to use more human tissue" (31; 00; 69) |
| | Franco 2018 [12] | "Having results from animal studies makes it easier to publish research in a high-ranking journal" (54; 32; 14) |
| | Van Luijk 2013 [23] | "3R implementation will lead to higher appreciation by journals" (27; 40; 33) |

(*Continued*)

**Table 2.** (Continued)

| Survey item | Publications included | Original survey item (%agreement; %neutral; %disagreement) |
|---|---|---|
| Q21: 3R implementation increases research costs | Bressers 2019 [16] | "Costs of implementation is a roadblock for implementation and development of alternatives" (68; 07; 25) |
| | Franco 2018 [12] | "In my case, replacing animal experiments for non-animal alternatives would be too expensive" (05; 61; 34) |
| | Lindsjö 2021 [19] | "Implementation of the 3R at your university results in increased research costs" (20; 38; 42) |
| | Nøhr 2016 [20] | "Alternative methods will increase research costs" (07; 55; 38) |
| | Van Luijk 2013 [23] | "3R implementation will increase research costs" (13; 27; 60) |
| Q22: 3R implementation increases bureaucracy | Bressers 2019 [16] | "Pressure to conform is a roadblock for implementation and development of alternatives" (60; 07; 33) |
| | Edwards 2015 [17] | "The regulatory requirements for the removal, storage and use of human tissues is a barrier to the use of human tissue" (43; 00; 57) |
| | Van Luijk 2013 [23] | "The obligation for 3R implementation increases bureaucracy" (27; 40; 33) |
| Q23: 3R implementation slows down innovation | Bressers 2019 [16] | "Time/effort to develop alternatives is a roadblock for implementation and development of alternatives" (74; 05; 21) |
| | Edwards 2015 [17] | "Better access to, and wider use of fresh human tissue would speed up the development of efficacious new therapies" (02; 10; 88) |
| | Lindsjö 2021 [19] | "Implementation of the 3R at your university results in slowed down innovation" (08; 44; 48) |
| | Nøhr 2016 [20] | "The extensive focus on the welfare of laboratory animals will hinder scientific breakthroughs" (07; 15; 78) |
| | Van Luijk 2013 [23] | "Application of the 3Rs slows down innovation" (00; 20; 80) |
| Q24: 3R implementation can have a bad impact on the career | Bressers 2019 [16] | "Would you consider migrating due to stricter governmental legislation" (46; 23; 31) |
| Q25: 3R implementation leads to non-comparable data | Bressers 2019 [16] | "Reliability is a roadblock for implementation and development of alternatives" (89; 05; 06) |
| | Edwards 2015 [17] | "Difficulty in predicting activity in isolated tissue compared to the complex integrated in vivo situation is a barrier to the use of human tissue" (25; 00; 75) |
| | Franco 2018 [12] | "'Refinement' measures can negatively interfere with the reproducibility of animal experiments" (12; 36; 52) |
| | Lindsjö 2021 [19] | "Researchers are reluctant to change the way they work because of the need for comparability with earlier findings" (32; 38; 30) |
| | Nøhr 2016 [20] | "I am reluctant to change the way I work because of the need for comparability with earlier findings" (17; 21; 62) |
| | Van Luijk 2013 [23] | "3R implementation needs to be rejected because of the necessity to compare results with earlier findings" (00; 07; 93) |

Furthermore, there is a perceived hierarchy of value between refinement efforts supported by scientific evidence and those proposed by animal technologists, which are deemed less legitimate by researchers. [26]. In situations where there was a disagreement between researchers and animal technologists/caretakers on the choice of the refinement method, researchers often held a more influential role in decision-making. Indeed, veterinarians–oftentimes acting as arbitrators–tended to give more academic credence to the decisions made by researchers when resolving conflicts [30]. For some authors, these power imbalances may foster a culture of separation and suspicion between generations and working sectors, impeding collaboration and mutual understanding [30].

Another reported systemic dynamic concerned the way of doing science and what is considered a "good" scientific career. Some researchers are used to experimenting on certain animal models and do not see the value of changing their practice [27,29]. Some others express

**Table 3. Overview of qualitative codes and themes included for meta-synthesis.**

| | Theme | Code | Publications included |
|---|---|---|---|
| **BARRIERS** | Systemic dynamics within animal experimentation | Publishing bias | Brønstad 2019 [24] |
| | | Lack of regulatory uptake | Brønstad 2019 [24] |
| | | Career incentives | Crettaz von Roten 2018 [22] |
| | | Scientific inertia | Mazhary 2019 [27] |
| | | Availability of animal models | Veening-Griffioen 2020 [29] |
| | | Power imbalances | Greenhough 2018 [26] Williams 2021 [30] |
| | | Tradition in animal science | Brønstad 2019 [24] Crettaz von Roten 2018 [22] Mazhary 2019 [27] |
| | Reification process of laboratory animals | Lack of respect for animal interests | Brønstad 2019 [24] Crettaz von Roten 2018 [22] Greenhough 2018 [26] Message 2019 [28] |
| | | Commodification of animals | Cabaret 2022 [25] |
| | | Professional persona | Cabaret 2022 [25] Williams 2021 [30] |
| | | Lifespan of laboratory animals | Cabaret 2022 [25] |
| | | Perceptions of animal behavior and anthropomorphism | Cabaret 2022 [25] Mazhary 2019 [27] Message 2019 [28] |
| | | Technology replacing the caring role of animal technologists | Greenhough 2018 [26] |
| | Practical issues | Financial constraints | Mazhary 2019 [27] |
| | | Lack of human resources | Williams 2021 [30] |
| | | Material resources | Brønstad 2019 [24] |
| | Insufficient knowledge and understanding of 3R | Misuse of reduction | Brønstad 2019 [24] |
| | | Prioritizing refinement over reduction | Brønstad 2019 [24] |
| | | Mismatch of refinement priorities | Message 2019 [28] |
| | | Limited knowledge of scientists on 3R | van Luijk 2013 [23] |
| **FACILITATORS** | Supportive workplace culture | Peer support | Brønstad 2019 [24] van Luijk 2013 [23] Veening-Griffioen 2020 [29] |
| | | Appreciation of animal caretakers/technologists' value | Cabaret 2022 [25] Greenhough 2018 [26] Mazhary 2019 [27] |
| | | Size of organization | Mazhary 2019 [27] |
| | | Collaboration | Greenhough 2018 [26] Mazhary 2019 [27] van Luijk 2013 [23] |
| | | Availability of tutors | Crettaz von Roten 2018 [22] |
| | | Training | Crettaz von Roten 2018 [22] Greenhough 2018 [26] van Luijk 2013 [23] Williams 2021 [30] |
| | More efficient use of 3R information and animal-based resources | Convincing evidence | Mazhary 2019 [27] |
| | | Awareness of available 3R methods | van Luijk 2013 [23] |
| | | Doing good science | Brønstad 2019 [24] Cabaret 2022 [25] Veening-Griffioen 2020 [29] |
| | | Re-use of existing data | Brønstad 2019 [24] |
| | | Sharing animal tissues and organs | Brønstad 2019 [24] Cabaret 2022 [25] |
| | | Literature search | Brønstad 2019 [24] van Luijk 2013 [23] Veening-Griffioen 2020 [29] |

(*Continued*)

**Table 3.** (Continued)

| | Theme | Code | Publications included |
|---|---|---|---|
| | Fostering animal care to empower change | Human-animal bond | Cabaret 2022 [25] <br> Williams 2021 [30] |
| | | Motivated staff | Mazhary 2019 [27] <br> Message 2019 [28] <br> van Luijk 2013 [23] <br> Williams 2021 [30] |
| | Regulation driving 3R forward | Regulatory compliance | Message 2019 [28] <br> Williams 2021 [30] |
| | | Ethics committees and animal welfare officers as facilitators | Cabaret 2022 [25] <br> Greenhough 2018 [26] <br> van Luijk 2013 [23] |

concerns about the feasibility of a transition to alternative methods without adequate alternatives available or a legal framework allowing their use [24,27]. Lastly, researchers describe pressure to use animal models and to publish in prestigious journals (which are reported to be less inclined towards alternative methods) in order to meet the criteria for a successful scientific career [22,24].

**2.4.3. Barrier 2: Reification process of laboratory animals.** Researchers frequently underscore the imperative of maintaining emotional detachment when working with laboratory animals, viewing it as a hallmark of professionalism and a prerequisite for upholding scientific objectivity [25,30]. This detachment is also described as a process for coping with animal deaths and is often considered as a rite of passage on the path to becoming a scientist [30]. While some applaud this emotional distance for its pragmatic contribution to scientific rigor [25], others critique it as eroding genuine care and attentiveness, *e.g.* when regulations mandate euthanasia of healthy animals [22,24–26]. Moreover, the increasing use of technology may isolate even more animal technologists from direct care work, limiting their ability to develop crucial skills and sensitivities required for innovative husbandry and colony management practices [26].

The capacity to objectify laboratory animals, by regarding them as research instruments rather than unique individuals, is deemed necessary but may act as a double-edged sword. Indeed, some publications report a dual detachment, where researchers appear unresponsive to both the animals' needs and the concerns of animal technologists and other workers responsible for their welfare [27,28,30]. These conversations surrounding the reification process extend to considerations specific to each species, with certain animals generating more emotional attachment and ethical concern than others [25,28]. Therefore, the sentiments and attitudes toward laboratory animals are influenced by factors such as species, exposure levels, and broader societal perceptions of animal sentience [28].

**2.4.4. Barrier 3: Practical issues.** Practical issues were reported only for refinement in the included publications, and several were recognized as barriers to it. In particular, financial concerns require integration of refinement costs into grant applications and encouragement of better budgeting for animal welfare [27]. Limited staff resources are also cited as a challenge, as staffing shortages hinders care standards [30]. But despite these constraints, a commitment to improving animal welfare seem to prevail among researchers, which suggests that if welfare benefits are made evident, financial barriers can be surmounted (*e.g.* by increasing public and private research funding) [24,27]. Other practical concerns worth mentioning are space constraints, which are typically considered unsolvable by most scientists, while some advocate for

refining scientific models to reduce animal numbers and maintain high-level science rather than mitigating welfare standards or scientific objectives [27].

**2.4.5. Barrier 4: Insufficient knowledge and understanding of 3R.** Researchers and facility managers often express uncertainties in applying reduction and refinement, resulting in unresolved dilemmas between achieving reduction and having enough animals to obtain reliable results, or in refining housing conditions [24]. Moreover, doubts are raised about the effectiveness and welfare benefits of structural enrichment, indicating a lack of clarity and scientific evidence regarding the refinement aspect of 3R [28]. Lack of awareness, knowledge, and educational resources to apply the 3R are also emphasized by Animal Welfare Officers [23].

**2.4.6. Facilitator 1: Supportive workplace culture.** A supportive workplace culture was a recurring theme throughout our qualitative analysis. Collaboration and cooperation were identified as facilitators [23] with open communication that facilitates raising issues [26], sharing ideas publicly [23], and sharing information or expertise on best practices [24,29]. This may be extended to communication between organizations in order to share ideas and to collaborate on projects, thus saving resources [27]. Supportive management [23] and colleagues [24] were identified as key facilitators especially for refinement and promoting good treatment of animals. It has been suggested that smaller organizations are better able to achieve this communicative culture than larger facilities [27].

Hands-on training and practice are seen as essential means to increase skills and to understand animal behaviors [22]. For some it is a mean of instilling pride in their work [30] and establishing the connection between animal welfare and good science. Further training to improve experimental design amongst researchers and those who oversee animal projects is recommended [23]. The standardization of required training (particularly for animal technologists) has improved regulatory compliance in terms of implementing 3R [26]. It follows that availability of experienced tutors with knowledge of up-to-date regulations and techniques is necessary in order to provide this level of essential training [22].

Another aspect of workplace culture affecting 3R implementation is how animal technologists are appreciated. Researchers are becoming more aware of their practical, in depth knowledge of animal husbandry [25]. With this understanding, animal technologists are able to identify impediments to animal welfare such as inadequate care or problematic breeding schemes [26]. Animal technologists can be instrumental in implementing and assessing the viability of new or modified practices, and they should therefore be given sufficient scope not only to raise concerns but also to suggest improvements [27].

**2.4.7. Facilitator 2: More efficient use of 3R information and animal-based resources.** Optimizing experimental design is associated with the refinement and reduction principles [25,29] and is revered as a mean of obtaining valid and reproducible results [24]. It is, for example, seen as crucial that animal stress is kept at a minimum to obviate it as a confounding variable in experiments [24]. Animal use is only considered justified when it occurs within a properly designed study [24] which involves the use of a so-called "good question" (or scientifically valid question) and appropriate means of addressing it [25]. Data from prior animal studies that successfully incorporated 3Rs measures are also an effective driver of change. They provide scientists with convincing 3Rs practices that are beneficial to either the animals they use or to their experimental setting [27], and the dissemination of these data should be prioritized at an organizational level using qualified appointed staff [23].

Along similar lines, scientists can make use of the already available resources for 3R implementation with efficient literature searches, in particular for replacement methods [29] and refinement techniques [24]. Although a centralized information system would be welcomed [23]. Existing data can be a valuable source of information and may avoid the need for some

animal use [24]. Further reduction could result from better sharing of organs and tissues between research projects [24]. This may involve harvesting several organs from one animal to be shared between several projects [25].

**2.4.8. Facilitator 3: Fostering animal care to empower change.**   Fostering a genuine concern for animal research subjects is deemed essential to provide optimum animal care [30]. Positive attitudes towards 3R, as well as an enthusiastic and compassionate staff, act as strong catalysts for implementing 3R practices [23]. This ability to care for animals may vary based on factors such as species, the nature of the procedures involved, and the duration of interaction with the animals [25]. Particularly crucial in the context of refinement, these humane resources play a pivotal role in shaping the unique bond between caregivers and animals. Animal technologists in particular value this bond, perceiving their role as advocates for the welfare of animals [30]. The capacity to care transforms individuals into motivated agents of change, empowering them to raise relevant issues, push for modifications [27], and import ideas from diverse fields to enhance and promote 3R techniques [28].

**2.4.9. Facilitator 4: Regulations driving 3R forward.**   The existence of ethics committees and compulsory regulations is seen as helpful by animal technologists and scientists as a means of clarifying what is permissible [25] and as a source of advice on optimal 3R implementation [23]. Inspectors, especially those with a veterinary background, are valued for their input in refinement [26]. Regulation updates can be an agent of change as they may result in a mindset within institutions that promotes animal welfare [30] and have the power to enforce improvements in everyday practice [28].

## 3. Discussion

### 3.1. General discussion

This systematic review identified 17 empirical publications that investigated how the animal research community in some European countries perceives the implementation of 3R. A major finding is that most people from this community do not believe that the "final goal of full replacement", as stated by the EU Directive 63/2010, is readily achievable (Q7 from meta-analysis). When they are asked about their own field of study, researchers do not see replacement as being possible (Q8 from meta-analysis). This is consistent with the conclusions of other commentators and shows the pressing need for more funding and regulatory uptake in replacement methods [4,13,31,32]. Furthermore, this review shed light on two important levers for 3R implementation: better dissemination of knowledge and the establishment of a culture of care in animal research.

### 3.2. Consolidating the regulatory uptake of 3R

The divergence between the perceptions of scientists on animal experimentation and the explicit goals set by regulatory bodies, such as the European Commission's vision of full replacement, is quite notable (71% in our meta-analysis), but it was also expected. Indeed, there is a deeply-rooted belief in the indispensability of animals as research models, and many scientists express skepticism about achieving even partial replacement of animal experiments, citing technological limitations, the need for whole-body system studies, and legal requirements to explain the need for *in vivo* research [10,19,20]. This impacts the prioritization of replacement by putting more emphasis on reduction and especially refinement methodologies, reversing the original order in Russell and Burch ([12] coined the term "upturned hierarchy" of the 3Rs). Thus, for many scientists, implementing the 3Rs actually implies the recognition of full refinement as a higher priority and a more readily achievable goal than replacement [11,12,33]. This matter was especially visible in our qualitative results, where only the practical

issues in relation to refinement were reported in the included publications, with no mention of reduction or replacement.

In order to promote the full implementation of 3R, we expect the regulatory framework to play a pivotal role in influencing the adoption of replacement methodologies, primarily through funding mechanisms. Our meta-analysis was not able to show whether or not a dedicated budget is perceived as beneficial for 3R implementation (Q6), but few scientists (9%) seem to believe that 3R implementation will increase their research costs (Q21). This matter was also not evident from our meta-synthesis, but it can still be argued that adequate financial support is likely to act as a crucial factor stimulating the development and implementation of new alternatives to animal experiments and 3R methods [33,34]. The prevailing grant strategy tends to view these alternatives as merely complementary rather than central to research programs. A shift towards dedicated centers with specific research programs based on alternative methods requires a reevaluation of funding strategies. Raising awareness about reliable open-access databases that document the available alternative methods, such as the European Union Reference Laboratory for Alternatives to Animal Testing (EURL-ECVAM) dataset on alternative methods to animal experimentation [35], can enhance funders' knowledge and encourage their use in laboratories.

On another note, more strategic planning of animal research is required. For instance, setting specific objectives both in terms of time and implementation level, *e.g.* as done by the Dutch "2025-goal" [16], may help to better engage communication between the government, researchers and the public, and to smooth active implementation of the 3Rs. In this regard, consolidating the existing legal apparatus by reinforcing the role of organizational and institutional culture should be prioritized, as some authors suggest that they are more influential than legal demands in achieving 3R improvements [36]. Indeed, institutional policies and laboratory culture regarding the ethical treatment of animals have a daily impact on researchers' awareness of animal welfare issues. A collaborative approach between researchers and 3R advisors is believed to be more efficient in promoting compliance with 3R recommendations than a paternalistic relationship with regulatory bodies [36]. To this end, the national 3R centers are an important feature, as they are able to coordinate and disseminate their resources, support and knowledge across research institutions [37].

### 3.3. Increasing knowledge about 3R

The meta-synthesis revealed insufficient knowledge and understanding of the 3Rs. At the same time, scientists acknowledge the need for more evidence in 3R methods and more efficient use of animal resources, which could be facilitated by dedicated animal specialists and knowledge transfer amongst institutions (Q16 and Q19 from meta-analysis). The promotion of data sharing should be seen as a catalyst that facilitates knowledge acquisition and prevent the unnecessary duplication of experiments. Indeed, the current reluctance to share research data prior to publication is problematic from a 3R perspective, as it leads to the replication of experiments and a potential waste of valuable resources (*e.g.* funding, laboratory devices and consumables, organs, unnecessary breeding). A cultural shift towards open data sharing may not only enhance insights within the research community, but it may also reduce the total number of sacrificed animals. The emergence of preregistration for animal studies [38], guidelines such as PREPARE [39] or ARRIVE [40], and the dissemination of the FAIR data principles [41] will certainly help to achieve this goal.

Our meta-analysis showed that 91% of scientists saw education as beneficial for 3Rs implementation (Q18). This result emphasizes that improved knowledge of replacement, reduction and refinement, as well as on animal research ethics, are essential for advancing the 3Rs. There

is a need for increased training and supervision of all stakeholders in order to enhance awareness and understanding of the 3Rs across the research community [42]. Therefore, the role of undergraduate and postgraduate training, in particular in universities, is paramount in educating scientists about the scientific validity of animal research and alternative methodologies to animal experimentation [32]. To this end, promoting initiatives such as the 3V principles on scientific validity (Construct Validity, Internal Validity, External Validity) [43] may further enrich the discourse by highlighting the interconnectedness of scientific quality and ethical considerations for effective and humane experimental techniques. Institutions can proactively offer courses on the development and implementation of 3R methods, fostering a culture of education among students and employees. This not only raises awareness of alternatives, but also promotes their consideration for experimental designs, contributing to a paradigm shift in research methodologies. This may in turn benefit scientific publication and the peer-review process, as it has been suggested that they are prone to conservationism bias and bias towards animal methods [44]. Moreover, Animal Welfare Bodies, which have been put into place along with the Directive 2010/63, can play an important role in laboratories, as they ensure the proper application of the 3Rs within research institutions. Encouraging a collaborative approach between scientists and Animal Welfare Bodies (only if the latter are properly resourced and supported by the institutional management) may foster a culture in which the 3Rs become the governing values, with constructive and neutral assistance offered by experts when problems are identified.

### 3.4. Improving adherence to 3R through a culture of care in animal research

The multifaceted nature of these findings also underscores the systemic challenges that hinder 3R implementation in animal research, emphasizing the need for a more equitable and collaborative approach. A majority of scientists have been shown to express occasional ethical concerns and dilemmas with regard to animal experimentation [12,45]. The bond formed between scientists and animals, particularly those deemed intelligent and receptive (*e.g.* mammals in comparison to fish), and those that have a longer life in the laboratory (*e.g.* farm animals in comparison to mice or rats), goes beyond mere utilitarian considerations. This connection may elicit compassion but can also result in compassion fatigue, especially among animal caretakers, who face the continuous challenge of balancing care with ethically challenging situations, including euthanasia [46,47]. This is also true for researchers, especially those in junior positions such as PhD students, who are more likely to report professional dissatisfaction or less social support [47,48]. Overall, there is increasing evidence that the wellbeing of laboratory animal professionals is negatively impacted by their work [45,47,49].

Instilling a workplace culture that addresses both animal welfare and care for the staff is a hallmark of a "culture of care", which has been gaining momentum for the past decade [50,51]. The European Federation of Pharmaceutical Industries and Associations' Research and Animal Welfare group members identified five areas for the successful implementation of a culture of care in animal research [52]: company values (including the celebration of 3R and care excellence), a strategic approach at the level of the institution (including training and mentorship), implementation structures (including the dissemination of good practices), staff support (including increased communication between researchers and animal care staff), and animal care and procedures (including animal supply and severity of experiments).

The recognition and acknowledgment of the intrinsic value of animals can mitigate the reification process, *i.e.* the tendency to view animals as mere research tools. Viewing animals as patients rather than statistical units, as suggested by some authors [53,54], may minimize

harm and underscore this individual worth. The shift towards a culture of care is especially relevant for animal caretakers who have historically faced a lack of recognition within research institutions, potentially leading to their own emotional distress. Building compassion satisfaction, i.e. the positive feeling and sense of fulfillment in providing care, and allowing more engagements between the animals and the research staff have been shown to reduce the occurrence of compassion fatigue in the laboratory setting [55–57].

The interconnection between research staff care, animal welfare and scientific practices challenges the dichotomy between care and scientific objectivity. Indeed, a culture of care may help to yield more reproducibility and improved quality in animal research, rather than compromising scientific excellence [51,58]. This is also one of the core messages from Russell and Burch when they wrote about the 3Rs:

> "Where chronic experiments over days or months are concerned, we cannot even in principle separate husbandry from the conduct of the experiment itself. [. . .] This is why the contribution of animal technicians is so important for the progress of humane experimentation, even when they do not themselves carry out actual experimental procedures such as the administration of drugs."

[1].

### 3.5. Strengths and limitations of this review

This review gives new insights into how the 3Rs have been implemented in Europe over the past ten years. The combination of both quantitative and qualitative analyses, whose findings are consistent with one another, as well as the diversity of countries assessed allows for a fair level of confidence in the conclusions that have been drawn. Reflexivity concerning the qualitative data, and researcher biases were mitigated through the use of validated tools and methodologies. There are also some limitations that may affect the results this review. First of all, the systematic search may have missed some relevant data, either because only three databases have been searched, because of the search strategy itself, because the publication languages were limited to English and French, or because only published studies were included. Secondly, the focus on the European context makes it difficult to translate the conclusions to other countries and contexts. Thirdly, the small amount of publications with quantitative data weakens the interpretation of the summary effects reported.

### 3.6. Implications for practice and recommendations for future research

To our knowledge, this is the first systematic review and synthesis of the existing literature on attitudes and perceptions of the animal research community on the implementation of 3Rs. This review shows first of all the limited number of studies dedicated to understanding of scientist's views on the 3Rs. The review provides evidence that actions should be taken both at a systemic level (the way of thinking about science and actually doing it) and at more local levels, such as research institutions (increase in 3R education) and interpersonal relationships (through the concept of culture of care, which includes both research staff and animal research subjects). In particular, the regulatory framework serves as a cornerstone to drive 3Rs implementation, influencing funding strategies, documentation practices, communication channels, and institutional cultures. A harmonized, more holistic, and more strategic effort involving national and international regulatory bodies, the animal research community and the public is therefore essential to achieve the full potential of the 3R principles.

As this review was focused on the European context, more research is needed in order to show whether these conclusions also apply in other countries. Future qualitative research is also needed in order to investigate how these findings are applicable in practice from the view-point of different stakeholders (*e.g.* researchers, animal welfare officers, animal technologists, policymakers) and in the different domains of animal experimentation (basic research, translational research, testing and education).

## 4. Materials and methods

### 4.1. Protocol

This review was conducted following the Preferred Reporting Items for Systematic Reviews and Meta-Analyses guidelines (PRISMA [14]). The PRISMA 2020 item checklist for this review can be accessed in S1 Appendix. The protocol was registered into the PROSPERO data-base under the number CRD42023395769. The literature search strategy included three online databases: PubMed, Web of Science (all collections), and Scopus. The complete search strategy for each database was reviewed by an information specialist from the University of Basel library and is reported in S2 Appendix. The final search was made on February 7th 2023 and targeted the title and abstract levels in order to maximize the relevance of the results.

### 4.2. Inclusion criteria

For feasibility reasons, due to time and resource constraints, the geographical locations were limited to all European countries that are known to have engaged in 3R policies, defined by countries having established a center dedicated to 3R implementation (a list has been provided by [59], and France, which has a 3R center since late 2021, has also been included). A specific timeframe (2013–2023) was also included, which is aligned with the intended implementation of the European Directive 2010/63/EU on animal experimentation into national laws by January 1st 2013 [9, p. 51].

For a publication to be included, the following criteria were thus applied: (a) the publication addresses the implementation of 3Rs in animal experimentation, (b) the publication includes experiences or perceptions from scientists, including researchers, animal caretakers, animal technologists, veterinarians, animal welfare officers and science students, (c) the publication originated in the United Kingdom, Austria, Belgium, Czech Republic, Denmark, France, Germany, Ireland, Italy, Luxembourg, the Netherlands, Norway, Poland, Portugal, Romania, Slovak Republic, Spain, Sweden, Switzerland, or Ukraine, (d) the study followed an empirical methodology, (e) the publication is either English or French, and (f) the publication is published between January 1st 2013 and January 1st 2023.

A publication was excluded if at least one of the following criteria was met: (a') absence of empirical data, (b') the location of the study includes a country that is not mentioned above, and (c') absence of experiences or perception from the scientific community.

### 4.3. Screening process

The literature search results were uploaded on EndNote X9, a reference manager, which was used for the automatic and manual removal of duplicates and subsequent organization and sharing of publications between review authors. In addition, the online software Covidence (https://www.covidence.org/) was used to consistently manage search results uploading and title-and-abstract as well as full-text screening.

The publications were independently screened at the level of the title and abstract by two review authors (ELM and AM). Then, the full texts of the relevant articles were obtained and

screened for inclusion by the same two review authors. If a discrepancy appeared at any of these stages, a third review author played the role of an arbitrator (LDG). Finally, the reference lists of all included publications were screened by a single review author (ELM) in order to identify other publications that satisfy the inclusion criteria.

## 4.4. Data extraction

A standard data extraction form was created and pilot-tested on ten included publications to ensure reliability and comprehensiveness of the data extraction process. Data were extracted independently by two review authors (ELM and AM). In case of discrepancies, a conciliation process was performed with further discussion on the data. Data extracted included: (a) general information about the publication, (b) information about the study design and objective, (c) quantitative data on 25 survey items related to 3R perception, 3R facilitators and 3R barriers, (d) qualitative data on perceived barriers and facilitators for 3R implementation. The 25 quantitative survey items were selected by the review authors based upon the topics raised in the publications retrieved through the preliminary search. Each survey item was divided into three categorical variables (namely "Agree" when respondents approved of the item, "Disagree" when respondents disapproved of the item, and "Neutral" when respondents did not answer the item or had a neutral opinion on the topic) and represented by proportions. As one of the objectives of this review was to provide a synthesis of the current literature, a meta-analysis and a meta-synthesis were conducted on the quantitative and qualitative data respectively.

## 4.5. Quantitative meta-analysis

A survey item was considered for meta-analysis if it appeared in at least two publications. This allows for the calculation of summary effects with reduced bias. Proportional meta-analysis was undertaken with the programming environment R (version 4.2.1). The full program used is reported in S3 Appendix. The proportion of agreement to the assessed survey item was systematically computed into the model. This implies that the negation actually includes both disagreements and a neutral opinion or an absence of opinion concerning the item. Because we hypothesized that the between-study heterogeneity was high (due to differences in time and place where the studies were conducted, their population, their specific context of study, or the exact formulation of each statement across studies), we performed a random effects model using an inverse variance approach with the logit transformation of proportions. The pooled effect estimates with 95% confidence intervals (95% CI) were presented along with prediction intervals and two measures of heterogeneity: between-study variance $\tau^2$ estimator (following Paule-Mandel method [60]), and $I^2$ statistics [61]. When the presence of outlying studies was suspected (defined as the 95% CI of their estimate not intersecting the 95% CI of the pooled effect estimate), an additional analysis excluding these studies was performed. Results were displayed using forest plots for each survey item.

## 4.6. Qualitative meta-synthesis

Qualitative data from the included publications (including direct quotes from study participants and their interpretations by study authors) were extracted and categorized as either perceived barriers or facilitators to 3R implementation. Three review authors (ELM, AM and LDG) conducted content analysis by inductively coding the data and grouping the different codes into broader themes. As emphasized by several methodologists [62–64], an iterative approach was followed in order to refine the coding and theming. The final themes were analyzed altogether using a narrative synthesis approach [65]. While doing so, a summary of the barriers and facilitators was developed, including an exploration of potential relationships. At

each stage of the synthesis, a collaborative attitude was adopted so that each review author was able to bring their own perspective on the data, thus enriching its overall quality.

### 4.7. Quality assessment

The Quality Assessment with Diverse Studies instrument (QuADS) was chosen in order to appraise the methodology and reporting of the included publications [66]. The QuADS tool is composed of thirteen items, each of them being scored from 0 (not satisfactory) to 3 (very satisfactory). Thus, the score for each publication can range from 0 to 39. After a pilot on three publications to discuss potential discrepancies, two review authors (ELM and AM) appraised the quality of each publication. The final result was expressed as a mean score among all review authors.

## Supporting information

**S1 Appendix. PRISMA checklist.**
(DOCX)

**S2 Appendix. Search strategy used for the systematic review.**
(DOCX)

**S3 Appendix. R program used for proportion meta-analysis.**
(DOCX)

**S4 Appendix. Proportion meta-analysis plots for all analyzed survey items.**
(PDF)

## Acknowledgments

The authors would like to thank the University of Basel information specialist team, in particular Dr. Christian Appenzeller-Herzog, for their esteemed support in designing the systematic search strategy.

## Author Contributions

**Conceptualization:** Edwin Louis-Maerten, Lester D. Geneviève.

**Data curation:** Edwin Louis-Maerten, Aoife Milford.

**Formal analysis:** Edwin Louis-Maerten, Aoife Milford, Lester D. Geneviève.

**Funding acquisition:** Bernice S. Elger.

**Investigation:** Edwin Louis-Maerten, Aoife Milford.

**Methodology:** Edwin Louis-Maerten, Aoife Milford, David M. Shaw, Lester D. Geneviève, Bernice S. Elger.

**Project administration:** Bernice S. Elger.

**Resources:** Bernice S. Elger.

**Software:** Edwin Louis-Maerten.

**Supervision:** David M. Shaw, Lester D. Geneviève, Bernice S. Elger.

**Validation:** David M. Shaw, Bernice S. Elger.

**Visualization:** Edwin Louis-Maerten.

**Writing – original draft:** Edwin Louis-Maerten, Aoife Milford.

**Writing – review & editing:** David M. Shaw, Lester D. Geneviève, Bernice S. Elger.

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
