## [Decision Letter · Decision Letter 0]

7 Feb 2024

PONE-D-23-42288Perceptions of 3R implementation in European animal research: A systematic review, meta-analysis, and meta-synthesis of barriers and facilitatorsPLOS ONE

Dear Dr. Louis-Maerten,

Thank you for submitting your manuscript to PLOS ONE. After careful consideration, we feel that it has merit but does not fully meet PLOS ONE’s publication criteria as it currently stands. Therefore, we invite you to submit a revised version of the manuscript that addresses the points raised during the review process.

The manuscript is very well designed so that it can be considered for publication in PloS One. Please consider the reviewer's comment.

We look forward to receiving your revised manuscript.

Kind regards,

Wesley Lyeverton Correia Ribeiro, Ph.D.

Academic Editor

PLOS ONE

Journal Requirements:

"The authors acknowledge the support of the Swiss National Science Foundation (SNSF National Research Program (NRP) - 79 Advancing 3R – Animals, Research and Society, grant number 206432) in funding the APC."

3. We note that this manuscript is a systematic review or meta-analysis; our author guidelines therefore require that you use PRISMA guidance to help improve reporting quality of this type of study. Please upload copies of the completed PRISMA checklist as Supporting Information with a file name “PRISMA checklist”.

Reviewers' comments:

Reviewer's Responses to Questions

**Comments to the Author**

1. Is the manuscript technically sound, and do the data support the conclusions?

Reviewer #1: Yes

2. Has the statistical analysis been performed appropriately and rigorously? 

Reviewer #1: Yes

3. Have the authors made all data underlying the findings in their manuscript fully available?

Reviewer #1: Yes

4. Is the manuscript presented in an intelligible fashion and written in standard English?

Reviewer #1: Yes

5. Review Comments to the Author

Reviewer #1: Dear Authors,

Congratulations on your work. I think it is very well designed, analysed and written. However, being a work oriented to the European Union, I am missing the work of a Spanish research group that has been working in this field in the last few years and that could be mentioned, since American groups are mentioned, it would not be superfluous to mention European groups. The works would be:

- PMID: 34315536 PMCID: PMC8314439 DOI: 10.1186/s42826-021-00098-w

- PMID: 34573605 PMCID: PMC8465412 DOI: 10.3390/ani11092639

- PMID: 35655241 PMCID: PMC9161537 DOI: 10.1186/s42826-022-00124-5

- PMID: 36496784 PMCID: PMC9735736 DOI: 10.3390/ani12233263

- PMID: 37684027 DOI: 10.1177/00236772231187177

This is just a recommendation, which I think would be interesting to be discussed in the discussion.

Best regards,

6. PLOS authors have the option to publish the peer review history of their article (what does this mean?). If published, this will include your full peer review and any attached files.

Reviewer #1: No

---

## [Author Response · Author response to Decision Letter 0]

14 Feb 2024

Dear Dr. Ribeiro,

We are grateful for your comments and the comments of the reviewer, which helped to improve the manuscript. Below we describe in detail how we responded to each comment.

Academic editor:

RESPONSE: The manuscript has been edited to meet the formatting requirements of the templates.

2. Thank you for stating the following financial disclosure: "The authors acknowledge the support of the Swiss National Science Foundation (SNSF National Research Program (NRP) - 79 Advancing 3R – Animals, Research and Society, grant number 206432) in funding the APC." Please state what role the funders took in the study. If the funders had no role, please state: "The funders had no role in study design, data collection and analysis, decision to publish, or preparation of the manuscript." If this statement is not correct you must amend it as needed. 

RESONSE: The financial disclosure statement has been amended accordingly in the manuscript. This statement has been added to the cover letter as well.

3. We note that this manuscript is a systematic review or meta-analysis; our author guidelines therefore require that you use PRISMA guidance to help improve reporting quality of this type of study. Please upload copies of the completed PRISMA checklist as Supporting Information with a file name “PRISMA checklist”.

RESPONSE: The PRISMA checklist is reported in the Supporting Information section as “S1 Appendix”. The filename has been amended.

RESPONSE: These references have been added to the reference list following Reviewer #1 suggestion:

Díez-Solinska A, Vegas O, Azkona G. Refinement in the European Union: A Systematic Review. Animals. 2022;12: 3263. doi:10.3390/ani12233263

Goñi-Balentziaga O, Vila S, Ortega-Saez I, Vegas O, Azkona G. Professional Quality of Life in Research Involving Laboratory Animals. Animals. 2021;11: 2639. doi: 10.3390/ani11092639

Goñi-Balentziaga O, Ortega-Saez I, Vila S, Azkona G. Working with laboratory rodents in Spain: a survey on welfare and wellbeing. Lab Anim Res. 2021;37. doi: 10.1186/s42826-021-00098-w

Goñi-Balentziaga O, Azkona G. Perceived professional quality of life and mental well-being among animal facility personnel in Spain. Laboratory Animals. 2023;0. doi:10.1177/00236772231187177

Reviewer #1:

Dear Authors,

Congratulations on your work. I think it is very well designed, analysed and written.

RESPONSE: Thank you very much for this positive feedback.

However, being a work oriented to the European Union, I am missing the work of a Spanish research group that has been working in this field in the last few years and that could be mentioned, since American groups are mentioned, it would not be superfluous to mention European groups. The works would be:

- PMID: 34315536 PMCID: PMC8314439 DOI: 10.1186/s42826-021-00098-w

- PMID: 34573605 PMCID: PMC8465412 DOI: 10.3390/ani11092639

- PMID: 35655241 PMCID: PMC9161537 DOI: 10.1186/s42826-022-00124-5

- PMID: 36496784 PMCID: PMC9735736 DOI: 10.3390/ani12233263

- PMID: 37684027 DOI: 10.1177/00236772231187177

This is just a recommendation, which I think would be interesting to be discussed in the discussion.

RESPONSE: Thank you for these suggestions. One of these references appeared in our systematic search, but could not be included in our analysis as it was not meeting all inclusion criteria. As the conclusions of some of these references relate to some aspects of our discussion, we included them where appropriate.

---

## [Editor Report · Decision Letter 1]

21 Feb 2024

Perceptions of 3R implementation in European animal research: A systematic review, meta-analysis, and meta-synthesis of barriers and facilitators

PONE-D-23-42288R1

Dear Dr. Louis-Maerten,

We’re pleased to inform you that your manuscript has been judged scientifically suitable for publication and will be formally accepted for publication once it meets all outstanding technical requirements.

Kind regards,

Wesley Lyeverton Correia Ribeiro, Ph.D.

Academic Editor

PLOS ONE
---

## [Editor Report · Acceptance letter]

19 Mar 2024

PONE-D-23-42288R1 

PLOS ONE

Dear Dr. Louis-Maerten, 

I'm pleased to inform you that your manuscript has been deemed suitable for publication in PLOS ONE. Congratulations! Your manuscript is now being handed over to our production team.

Kind regards, 

on behalf of

Dr. Wesley Lyeverton Correia Ribeiro 

Academic Editor

PLOS ONE